# Back to the Future: Rethinking the Great Potential of lncRNA_S_ for Optimizing Chemotherapeutic Response in Ovarian Cancer

**DOI:** 10.3390/cancers12092406

**Published:** 2020-08-25

**Authors:** Abdelrahman M. Elsayed, Paola Amero, Salama A. Salama, Abdelaziz H. Abdelaziz, Gabriel Lopez-Berestein, Cristian Rodriguez-Aguayo

**Affiliations:** 1Department of Experimental Therapeutics, The University of Texas MD Anderson Cancer Center, Houston, TX 77030, USA; AMHamoda@mdanderson.org (A.M.E.); PAmero@mdanderson.org (P.A.); glopez@mdanderson.org (G.L.-B.); 2Department of Pharmacology and Toxicology, Faculty of Pharmacy, Al-Azhar University, Cairo 11675, Egypt; salama@azhar.edu.eg (S.A.S.); abdelazizhamed@azhar.edu.eg (A.H.A.); 3Department of Cancer Biology, Center for RNA Interference and Non-Coding RNA, The University of Texas MD Anderson Cancer Center, Houston, TX 77030, USA

**Keywords:** long non-coding RNAs, mechanisms of resistance, chemotherapy, dysregulated expression, ovarian cancer

## Abstract

Ovarian cancer (OC) is one of the most fatal cancers in women worldwide. Currently, platinum- and taxane-based chemotherapy is the mainstay for the treatment of OC. Yet, the emergence of chemoresistance results in therapeutic failure and significant relapse despite a consistent rate of primary response. Emerging evidence substantiates the potential role of lncRNAs in determining the response to standard chemotherapy in OC. The objective of this narrative review is to provide an integrated, synthesized overview of the current state of knowledge regarding the role of lncRNAs in the emergence of resistance to platinum- and taxane-based chemotherapy in OC. In addition, we sought to develop conceptual frameworks for harnessing the therapeutic potential of lncRNAs in strategies aimed at enhancing the chemotherapy response of OC. Furthermore, we offered significant new perspectives and insights on the interplay between lncRNAs and the molecular circuitries implicated in chemoresistance to determine their impacts on therapeutic response. Although this review summarizes robust data concerning the involvement of lncRNAs in the emergence of acquired resistance to platinum- and taxane-based chemotherapy in OC, effective approaches for translating these lncRNAs into clinical practice warrant further investigation.

## 1. Introduction

Ovarian cancer (OC) is the most lethal gynecological malignancy in women, accounting for approximately 150,000 annual deaths worldwide. For patients with advanced stages of OC, overall survival averages only 5 years due to the disease’s asymptomatic nature during the early stages, leading to late diagnosis and resistance to conventional chemotherapy. Although most patients initially respond to chemotherapy, the majority of them relapse and develop resistance to chemotherapy [1]. Based on its genetic behavior and histological features, OC can be classified into two major types [2]. Type I includes mucinous, clear cell, endometrioid, and low-grade serous carcinomas, and mutations of *PTEN, BRAF, KRAS*, and *CTNNB1* have been identified [3]. Type II accounts for approximately 90% of OC cases and comprises high-grade serous cancers, carcinosarcomas, and undifferentiated carcinomas. Remarkably, type II tumors include more aggressive epithelial malignancies that have a higher tendency for invasion and metastasis. These tumors are associated with multiple mutations of *TP53* and *BRCA1/2* genes [4]. Notably, high-grade serous carcinoma is the most common, with the highest mortality rate [5].

Currently, the standard treatment approach for newly diagnosed OC patients includes surgical removal of the tumor followed by chemotherapy with a platinum-based antineoplastic (either cisplatin or carboplatin) combined with a taxane (either paclitaxel (PTX) or docetaxel) [6]. An alternative strategy comprises neoadjuvant chemotherapy followed by tumor debulking and subsequent chemotherapy. Tumor recurrence as a result of the emergence of platinum resistance is the most difficult issue for the management of OC [5]. Advances in OC management include the use of angiogenesis inhibitors such as bevacizumab, poly (ADP-ribose) polymerase (PARP) inhibitors, and immunotherapy-based agents. However, due to the development of chemoresistance, the outcomes of these new treatment modalities have been dismal as there was no significant increase in overall survival [5].

It was previously thought that most of the human genome, up to 75%, is transcribed into non-coding RNAs (ncRNAs), whereas less than 2% of the human genome encodes protein. However, recent data shed light on the numerous and indispensable functions of ncRNAs, particularly in carcinogenesis and chemoresistance [7,8,9]. Long non-coding RNAs (lncRNAs) are defined as RNA transcripts comprising more than 200 nucleotides, and most of them are incapable of encoding proteins [10]. Nevertheless, recent evidence has proven that lncRNAs can encode for peptides/proteins, which usually contain less than 100 amino acids [11,12]. According to the data obtained from the Encyclopedia of DNA Elements (ENCODE) project, more than 28,000 lncRNA transcripts are generated by approximately 16,000 genes and represent the most diverse class of ncRNAs [13]. In the context of cancer, mounting evidence indicates that aberrant expression of lncRNAs is associated with the development of multiple malignancies [14]. In OC, certain lncRNAs function as oncogenes, while others might function as tumor suppressors and therefore modulate tumor growth and response to chemotherapy [15]. Indeed, certain lncRNAs can be targeted as a potential therapeutic approach to ameliorate chemoresistance [15]. Thus, in this review, we summarize the potential mechanisms of lncRNAs underlying resistance to platinums and taxanes in OC.

## 2. Classification of lncRNAs

lncRNAs are broadly classified based on genomic origin into intergenic, intronic, bidirectional, sense, and antisense lncRNAs. Intergenic lncRNA originates from a genomic sequence located between two subsequent protein-coding genes [16]. Intronic lncRNA develops when the entire sequence of lncRNA is located within the intron of a protein-coding gene [17]. Bidirectional lncRNA exists within 1 kb of the protein-coding gene promoter, but is transcribed from the opposite DNA strand [18]. Sense and antisense lncRNAs develop if located between one or more exons of another transcript on the same or opposite strand, respectively (Figure 1) [19,20].

## 3. Localization of lncRNAs

It is well known that mRNA is first biosynthesized in the nucleus as a pre-mRNA then exported into the cytoplasm, where it can easily be recognized by ribosomal and transfer RNAs, eventually leading to protein translation. Unlike mRNA, the functions of lncRNAs rely mainly on physical interactions with either RNAs or proteins, and these interactions in turn depend on close vicinity; therefore, lncRNAs can find their way to particular locations inside or outside the nucleus [21]. Although the prevailing narratives reported that the nucleus harbors most lncRNAs where they can regulate gene expression mainly via epigenetic regulation, subsequent studies have identified many lncRNAs with cytoplasmic localization [22,23]. For instance, Bouvrette et al. demonstrated that approximately 75% of lncRNAs are found in cytoplasmic fractions of human and Drosophila cells [24]. Similarly, a higher proportion of lncRNAs (about 40%) has been identified in the cytoplasm compared with approximately 4% nuclear fraction in a study conducted via using RNA fluorescence in situ hybridization (FISH) and Drosophila models [25]. lncRNAs have a complex distribution in a wide variety of cellular compartments and organelles. Within the cytoplasm, lncRNAs can present in mitochondria [26], ribosomes [27], extracellular membranes [28], exosomes [29], and other components. In the nucleus, lncRNAs have been identified in chromatin speckles (nuclear domains enriched in splicing factors), paraspeckles (ribonucleoprotein domains located at interchromatin spaces and involved in nuclear retention of mRNA), and nucleoli (the largest body in the nucleus, where the ribosomal biosynthesis occurs) [23,30,31]. In conclusion, identifying lncRNAs’ subcellular localization may pave the path for determining their subsequent biological functions [23].

## 4. Major Biological Functions of lncRNAs

Although lncRNAs exert innumerable biological functions, their action can be generally categorized into four main archetypes of molecular mechanisms: signals, guides, decoys, and scaffolds. Remarkably, the function of lncRNAs is multifactorial in that a single lncRNA can fulfill multiple archetypes [32]. Given that lncRNA expression patterns are cell and tissue specific and that diverse stimuli can induce lncRNA expression, some lncRNAs can serve as molecular signals in response to particular stimuli, and these lncRNAs can be utilized as biomarkers in different biological events [32]. In addition, lncRNAs can also function as guide molecules, in which lncRNAs bind to particular regulatory proteins or enzymes like transcription factors or chromatin-modifying complexes and precisely direct them to the exact locations within the genome to regulate gene expression [33]. For instance, the homeobox transcript antisense RNA (HOTAIR), located at the HOXC locus, guides polycomb repressive complex 2 (PRC2) to the HOXD locus to regulate gene expression. Importantly, aberrant expression of HOTAIR is associated with the development of several types of cancer [34]. In contrast to the guide function, some lncRNAs can serve as decoys by binding to and sequestering miRNAs, RNA-binding proteins, catalytic proteins, modifying complexes, and transcription factors to negatively regulate their function [35]. In the case of miRNAs, certain lncRNAs regulate gene expression by sequestering miRNAs and circumventing the subsequent interaction with the target mRNAs and therefore inhibiting miRNA-induced gene silencing [36]. This mode of action is called “miRNA sponging,” and the lncRNAs that act by this mechanism are called competitive endogenous RNAs (ceRNAs) [37]. The fourth and the most intricate molecular archetype is the scaffold function. In this archetype, lncRNAs function as docking sites upon which distinct effector proteins interact at the same time, thereby bringing the effector molecules together at the same time and place. This scaffolding process enables the transmission of signals and therefore the regulation of multiple biological processes [32]. In addition, lncRNAs exert diverse regulatory functions that originate from these archetypes. For instance, lncRNAs regulate gene expression by regulating (1) chromatin remodeling; (2) transcription and post-transcription modifications, e.g., alternative splicing; (3) translation and post-translation modifications, e.g., ubiquitination and phosphorylation; and (4) mRNA and protein stability [35]. Finally, although originally identified as non-protein coding, recent studies have shown that some lncRNAs encode micropeptides with definitive biological functions (Figure 2) [38].

## 5. lncRNAs and Chemoresistance in Ovarian Cancer

The therapeutic approach for advanced OC relies on maximum tumor debulking followed by platinum- and taxane-based chemotherapy. Despite advances in therapeutic modalities, intrinsic or acquired resistance to platinum and/or taxane therapy remains the major reason for poor survival rates in OC patients, even when the initial response is promising [6]. Thus, understanding the molecular mechanisms implicated in chemoresistance is indispensable to improve the overall clinical outcomes of OC patients. A growing body of evidence suggests that aberrant expression of lncRNAs is associated with the development of multiple malignancies and chemoresistance [14]. Differential expression of lncRNAs has been identified in cisplatin-resistant and cisplatin-sensitive OC, advocating their predicted role in the emergence of chemoresistance. However, precisely how lncRNAs contribute to chemoresistance remains undefined. Here, we provide an updated insight into the molecular mechanisms by which lncRNAs induce pharmacological resistance and how they represent potential targets in the development of effective therapy and biomarkers in OC.

### 5.1. lncRNAs and Platinum Resistance

Since its approval by the US Food and Drug Administration in 1987, cisplatin and other platinum-based compounds have become the foundation for the management of a wide array of solid malignancies including ovarian, testicular, head and neck, lung, bladder, esophageal, and colon cancers. Currently, three platinum-based agents are used clinically: cisplatin, carboplatin, and oxaliplatin. Among those, only cisplatin and carboplatin are approved for the first-line treatment of advanced-stage OC [39]. Mechanistically, platinum analogues exert their action mainly through binding to the N7 position of guanine nucleotides, leading to the formation of DNA adducts, intrastrand and interstrand cross-links. The resultant single- and double-strand breaks impact DNA repair machinery, thereby inducing cell cycle arrest and apoptosis. In addition, platinum analogues enhance the release of reactive oxygen species (ROS), which in turn contribute to platinum-induced cytotoxicity [40]. Despite successful initial responses, many OC patients acquire chemoresistance to cisplatin and/or carboplatin, resulting in overall therapeutic failure [40]. Therefore, there is an urgent need to uncover the molecular pathways implicated in platinum resistance to improve therapeutic outcomes.

The mechanistic circuitries encountered in platinum resistance are multifactorial and include reduced intracellular drug uptake, enhanced drug efflux via multi-drug resistant transporters, and increased intracellular sequestration by nucleophilic scavengers such as reduced glutathione (GSH) and metallothioneins. Likewise, the activation of DNA repair systems, evasion of apoptosis, alteration of autophagy, increased cancer stemness, and activation of survival pathways such as the PI3K/AKT axis play a fundamental role in conferring resistance to platinum-based chemotherapy [9,41]. The activation of DNA repair systems such as the nucleotide excision repair and homologous recombination (HR) can retard the apoptotic process, ultimately leading to platinum resistance. Importantly, the downregulation of breast cancer susceptibility proteins 1 and 2 (BRCA1/2), the two critical factors in the homologous recombination system, enhances platinum sensitivity in multiple cancers [42]. The activation of the mismatch repair (MMR) system also enhances platinum sensitivity because it can recognize platinum-induced DNA lesion and consequently activate apoptosis. Importantly, the downregulation of MMR-related genes such as *MLH1* and *MSH2* is implicated in the emergence of platinum resistance [43]. Furthermore, dysregulation of tumor suppressor protein p53 and its related nuclear transcription factors play a fundamental role in platinum resistance [44,45]. Accumulating evidence indicates that several lncRNAs are associated with the emergence of cisplatin or carboplatin resistance. In Table 1, we present and categorize these lncRNAs based on the mechanisms implicated in platinum chemoresistance.

#### 5.1.1. Reduced Intracellular Accumulation

One of the major mechanisms encountered in platinum resistance is reduced intracellular drug accumulation as a result of either reduced uptake or enhanced efflux of the drug. Thus, the regulation of protein transporters that mediate cisplatin uptake and efflux plays a fundamental role in platinum resistance. Of those transporters, copper transporter 1 (CTR1), a transmembrane protein that regulates copper homeostasis, plays a significant role in cisplatin uptake across the cell membrane [40]. Interestingly, the downregulation of CTR1 significantly blocked cisplatin uptake in yeast and in mouse embryonic fibroblasts [71,72]. Zinc finger antisense 1 (ZFAS1) is an lncRNA transcribed from the antisense strand in close proximity to the protein-coding gene *ZNFX1* [73]. In cancer, ZFAS1 has a tissue-specific function as evidenced by its different expression patterns among various human malignancies [9]. In OC, ZFAS1 was associated with cisplatin resistance, and silencing ZFAS1 improved cisplatin sensitivity, at least in part, by regulating the miR-150-5p/specific protein 1 (SP1) axis [46]. SP1 is a transcriptional factor that regulates CTR1 [74] and DNA damage response (DDR) [75]. Thus, ZFAS1 could induce cisplatin resistance through modulating SP1-mediated CTR1 expression. More studies need to be conducted to reveal the exact relationship between lncRNAs and CTR1 in OC. In lung cancer, nuclear-enriched abundant transcript 1 (NEAT1) has been shown to enhance cisplatin sensitivity via upregulating epigallocatechin-3-gallate (EGCG)-induced CTR1 expression [49].

Previous reports have demonstrated the role of ATP-binding cassette (ABC) transporters, particularly P-glycoprotein and multi-drug resistant proteins (MRPs), in platinum resistance via triggering the efflux process [41]. Metastasis-associated lung adenocarcinoma transcript 1 (MALAT-1), the first identified cancer-associated lncRNA, is upregulated in malignancies such as ovarian, lung, colon, cervical, and prostate cancer and employed as a prognostic factor for metastasis and survival [76,77]. Bai et al. elucidated that MALAT1 is overexpressed in OC and is positively correlated with cisplatin resistance and Notch1 in OC tissues and cisplatin-resistant cells [47]. Mechanistically, silencing MALAT1 significantly inhibited Notch1 and ABCC1 and enhanced cisplatin chemosensitivity. Moreover, Notch1 knockdown reversed cisplatin resistance and reduced the expression of *ABCC1/MRP1* in cisplatin-resistant cells. Based on these data, the authors speculated that MALAT1 enhances cisplatin resistance through regulating the Notch1/ABCC1/MRP1 signaling pathway, and therefore MALAT 1 could be used as a promising therapeutic approach to enhance cisplatin sensitivity in OC (Figure 3) [47].

#### 5.1.2. Intracellular Detoxification

In the context of its mechanistic pathway, cisplatin not only interacts with purine nucleotides to activate DNA damage-induced apoptosis but also induces generation of ROS that participate in the platinum-induced cytotoxic effect [41,78]. Accordingly, regulation of the antioxidant system plays a substantial role in cisplatin resistance. Reduced GSH is a type of metallothionein that has a higher affinity to cisplatin than DNA [40,79]. Remarkably, elevated GSH level enhances cisplatin resistance, whereas depletion of GSH stores augments cisplatin chemosensitivity [80]. Moreover, upregulation of enzymes implicated in GSH synthesis and metabolism can mediate cisplatin resistance [50]. *H19,* also known as imprinted maternally expressed transcript, is an evolutionary conserved lncRNA expressed mainly in placental and fetal tissues, where it plays a fundamental role during embryogenesis [81]. It was reported that *H19* overexpression is positively correlated with cisplatin resistance in non-small cell lung cancer [82] and seminoma [83]. In OC, transcriptome sequencing elicited differential expression of *H19* in cisplatin-resistant cells compared with wild-type cells. Additionally, the involvement of *H19* in platinum resistance was evidenced in a study of tissues obtained from 41 high-grade serous carcinoma patients treated with either cisplatin or carboplatin; the results revealed that *H19* positively correlated with the early recurrence of OC [50]. Mechanistically characterized using proteomic analysis, it was shown that *H19* knockdown reduces the expression of six NRF2-targeted proteins: NQO1, GSR, G6PD, GCLC, GCLM, and GSTP1. Nuclear factor erythroid 2 (NRF2) is a transcription factor that regulates the expression of multiple antioxidant enzymes, particularly those involved in the GSH metabolic pathway. Notably, the glutathione pathway plays a pivotal role in platinum resistance via counteracting platinum-induced oxidative cytotoxicity [84]. *H19* is believed to enhance cisplatin resistance through activating GSH-dependent antioxidant activity, eventually abrogating cisplatin-induced oxidative cytotoxicity (Figure 3) [50].

#### 5.1.3. Regulation of Autophagy

Autophagy is a conserved cellular process in which the cells degrade and remove dispensable or dysfunctional cytoplasmic components by the aid of lysosomal compartments. Autophagy plays a fundamental role in maintaining energy homeostasis, particularly during development and starvation [85]. Recently, multiple studies demonstrated that autophagy induces cisplatin resistance in cancer cells [86,87]. Yu et al. reported that lncRNA HOTAIR is overexpressed in OC and associated with cisplatin resistance [52]. Mechanistically, HOTAIR induced autophagy and increased cisplatin resistance through activating ATG7 [52]. ATG7 regulates autophagosome assembly via binding to and activating ubiquitin-like proteins such as ATG12 and ATG8 [88]. Similarly, HOTAIR promoted tumor growth and exacerbated cisplatin resistance via activating the Beclin-1/autophagy axis in endometrial cancer (EC) [53]. The X-inactive specific transcript (XIST) is an lncRNA identified within the region of chromosome X inactivation center (XIC) that plays a fundamental role in silencing one X chromosome during the developmental process in female fetuses [89]. It was reported that XIST is upregulated in non-small cell lung cancer tissues and positively correlated with autophagy and cisplatin resistance [90]. Functional analysis revealed that XIST knockdown increased cisplatin chemosensitivity and suppressed autophagy, and the administration of miR-17 inhibitor or upregulation of *ATG7* reversed this action. These data proposed that XIST confers cisplatin chemoresistance, at least in part, through regulating the miR-17/ATG7 axis (Figure 3) [90].

#### 5.1.4. Epithelial–Mesenchymal Transition

Epithelial–mesenchymal transition (EMT) is a reversible cellular process whereby epithelial cells lose cell polarity and cell–cell adhesion to adopt high migratory, spindle-shaped mesenchymal stem cells. EMT is important for numerous biological processes such as embryogenesis, development, wound healing, and cancer progression [91]. In the context of tumor progression, EMT can gain multiple traits that are important for tumor initiation, motility, metastasis, and the emergence of chemoresistance. Importantly, the presence of EMT-associated proteins can be exploited as highly specific predictors of high-grade malignancy [91,92]. EMT is orchestrated by certain transcription factors, including zinc-finger E-box-binding homeobox factors 1 and 2 (ZEB1 and ZEB2), SNAIL, SLUG, TWIST1, and TWIST2 [93,94]. These transcription factors are further regulated by particular upstream signaling pathways, including TGF-β, HIF1-α, Wnt/β-catenin, and Notch, and the dysregulation of these signaling cascades is associated with the emergence of platinum resistance [9,95]. It was reported that silencing HOTAIR enhanced cisplatin sensitivity in a mouse xenograft model of OC [55]. Mechanistically, this effect could be attributed to the reduced activity of the Wnt/β-catenin-induced EMT [55]. As another example, overexpression of *H19* promoted tumor growth, migration, and cisplatin resistance in OC cells by triggering EMT, as indicated by reduced E-cadherin and increased TWIST, SLUG, and SNAIL mRNA and protein levels (Figure 3) [51].

#### 5.1.5. Repair of Damaged DNA

DDR comprises a network of multiple signaling pathways that orchestrate DNA repair in response to DNA damage to prevent the development of potentially harmful mutations [96]. The DDR signaling network generally includes the classic tumor suppressor gene *TP53*, its downstream target *p21*, and nuclear factor kappa B (NF-κB). In the setting of cancer, DDR activation promotes cancer cell survival and confers resistance to chemotherapy [96,97]. Since platinum compounds exert their major cytotoxic action by inducing DNA damage, cells that acquire high potential for repairing or preventing DNA damage can develop platinum resistance. Remarkably, DNA damage triggers the release of p53-dependent pro-apoptotic factors, and therefore decreased activity of p53 or its downstream signaling pathway can contribute to platinum resistance [98]. It was demonstrated that HOTAIR induces cisplatin resistance in OC via activating DDR. Mechanistically, NF-κB directly enhances HOTAIR expression in OC cell lines following cisplatin-induced-DNA damage. Subsequently, HOTAIR acts in a positive feedback loop to trigger persistent NF-κB activation, IL-6 secretion, and CHK1-p53-p21 pathway stimulation. Consequently, these effects drive the establishment of OC senescence and emergence of chemoresistance. The authors speculated that HOTAIR could be used as a new therapeutic strategy to improve platinum resistance in OC [56]. *GAS5*, also known as growth arrest-specific transcript 5, is a tumor suppressor gene located at chromosome 1q25 and encodes an lncRNA [99]. Dysregulation of GAS5 has been reported in various tumors, including lung [100], cervical [101], liver [102], and breast cancers [103]. In OC, GAS5 has been shown to inhibit tumor growth and inversely correlate with cisplatin resistance. In addition, GAS5 overexpression induced G0/G1 cell cycle arrest, apoptosis and cisplatin chemosensitivity, indicating its tumor suppressor effect. Mechanistic analysis revealed that GAS5 inhibits PARP1 expression via recruiting the E2F4 transcriptional factor to the promoter region of the *PARP1* gene [48]. In the context of biological function, PARP1 plays an important role in genomic stability via regulating DNA repair, cell cycle progression, the evasion of apoptosis, and the activation of the mitogen-activated protein kinase (MAPK) downstream signaling pathway [104]. As a consequence of PARP1 inhibition, GAS5 overexpression inhibited the phosphorylation of extracellular signal-regulated kinase (ERK), jun N-terminal kinase (JNK), and MAPK, suggesting that GAS5 inhibits tumor growth and abrogates cisplatin resistance via regulating the E2F4-PARP1-MAPK signaling pathway (Figure 4) [48].

#### 5.1.6. Regulation of Apoptosis

Apoptosis is an evolutionarily conserved process that occurs in both physiological and pathological conditions and results in cell death. Two main pathways of apoptosis have been identified, known as extrinsic and intrinsic [105]. The intrinsic or mitochondrial pathway is mainly regulated by the BCL-2 family that includes antiapoptotic proteins such as BCL-2 and BCL-XL and proapoptotic proteins such as BAX, BAK, BAD, and BIK [106]. Since platinum agents exert their action mainly by inducing DNA damage with subsequent activation of p53-induced apoptosis, the suppression of apoptosis accounts for the acquisition of platinum resistance [41]. Several reports indicated that certain lncRNAs inhibit p53-induced apoptosis leading to acquisition of resistance [41]. For example, the lncRNA PANDAR has been identified as the most overexpressed p53-dependent lncRNA regulated by drug-induced apoptosis [107]. PANDAR has been shown to augment cisplatin resistance via modulating the PANDAR/SFRS2/p53 feedback loop in the nucleus. Mechanistically, PANDAR-induced cisplatin resistance was mediated by the PANDAR-binding protein, SFRS2 (arginine/serine-rich 2). SFRS2 is a splicing factor that negatively regulates P53 and its phosphorylation at Serine 15 (Ser15) [108,109]. Accordingly, the PANDAR–SFRS2–p53 feedback axis markedly decreases the transactivation of p53-related pro-apoptotic genes and consequently triggers cisplatin chemoresistance [58]. E2F-mediated cell proliferation enhancing lncRNA (EPEL) was originally identified in lung cancer [110]. Zhao et al. reported that the plasma level of EPEL is elevated in endometrioid adenocarcinoma patients compared to healthy patients [59]. Mechanistically, the overexpression of EPEL markedly dampened the expression of p53, thereby conferring carboplatin resistance. Additionally, p53 overexpression did not alter EPEL expression level, suggesting that EPEL is an upstream inhibitor of p53 [59]. In a similar way, a study has been conducted to determine the effect of the carboplatin/docetaxel combination on the expression levels of several lncRNAs in an OC cell line, 3AO [60]. Among 30 tumor-related lncRNAs, only 10 different lncRNAs showed significant alterations in response to carboplatin/docetaxel treatment. These 10 lncRNAs include PVT1, TDRG1, PCAT1, GAS5, HOTAIR, SRA1, BCYRN1, CASC2, HI9, and MEG3. The first six lncRNAs were upregulated while the last four were downregulated following carboplatin/docetaxel treatment. The six upregulated lncRNAs were silenced by using their specific siRNAs, and only siRNA/PVT1 showed increased cell proliferation when combined with docetaxel/carboplatin treatment, indicating that carboplatin/docetaxel combination induces PVT1 expression, which in turn could mediate the anticancer activity of the combined treatment. On a mechanistic level, PVT1 enhanced the expression level of both TIMP1 and p53. TIMP1, also called tissue inhibitor of matrix metalloproteinase 1, inhibits the proteolytic activity of matrix metalloproteinases, eventually leading to the inhibition of tumor invasion and metastasis [111,112], whereas p53 regulates the cell cycle and thus functions as a tumor suppressor protein. Through upregulating TIMP1 and p53, lncRNA PVT1 improves sensitivity to carboplatin/docetaxel treatment and exerts direct tumor-suppressing action [60].

Colon cancer-associated transcript 1 (CCAT1) is an lncRNA mapped at chromosome 8q24.21 and has been initially identified as an oncogenic lncRNA in colorectal cancer [113]. Subsequent studies elucidated that CCAT1 is overexpressed not only in colorectal cancer but also in various types of cancers, such as liver, ovarian, lung, gastric, breast, and gallbladder cancers [114,115]. In OC, it was reported that CCAT1 is upregulated in cisplatin-resistant cells, A2780/CP and SKOV3/CP, and CCAT1 knockdown markedly enhanced cisplatin chemosensitivity via inducing apoptosis in vitro and in vivo [61]. Mechanistically, silencing CCAT1 significantly elevated the level of miR-454, decreased levels of anti-apoptotic proteins BCL-2 and survivin, and enhanced the level of pro-apoptotic protein BAX. Research using bioinformatics and dual luciferase reporter assay identified miR-454 as a direct target for CCAT1. This association was further proved through functional experiments in which miR-454 overexpression enhanced cisplatin cytotoxicity, whereas miR-454 inhibitors exacerbated CCAT1-induced cisplatin resistance—suggesting that CCAT1 enhances cisplatin resistance, at least in part, through modulating the miR-454/survivin axis [61]. In other research, LINC00152 was upregulated in cisplatin-resistant OC cells, and LINC00152 knockdown significantly abrogated cisplatin resistance via decreasing MDR1 and MRP1 mRNA levels and inducing apoptosis, as indicated by increased cleaved caspase 3 and BAX and reduced BCL-2 protein levels [62]. Other lncRNAs can affect cisplatin resistance through regulating apoptosis; however, the exact underlying mechanism for their action has not been revealed yet. For instance, PVT1 enhances [116], but ENST00000457645 reduces [66] cisplatin resistance in OC (Figure 4).

#### 5.1.7. Sponging miRNAs

Several lncRNAs regulate gene expression by sequestering miRNAs and acting as competing endogenous RNAs (ceRNAs), eventually abrogating miRNA-induced gene silencing [36]. Urothelial carcinoma associated 1 (UCA1) is an oncogenic lncRNA associated with multiple malignancies including hepatocellular carcinoma, ovarian, bladder, breast, and colorectal cancers [117,118,119,120,121]. In OC, it was demonstrated that UCA1 is upregulated in cisplatin-resistant cells and patient tissues. Further mechanistic studies revealed that UCA1 exacerbates cisplatin resistance via regulating the miR-143/FOSL2 axis. Notably, miR-143 is a tumor suppressor that negatively correlates with cisplatin resistance and is believed to exert its action, at least in part, through silencing FOSL2 gene expression [63]. Fos-related antigen 2 (FRA-2/FOSL2) belongs to the AP-1 transcription factor family and plays a critical role in tumor growth and metastasis [122,123,124]. Dual-luciferase reporter and RIP assays identified the direct interaction between UCA1 and miR-143, indicating that UCA1 induces its effect via sponging miR-143 with a subsequent upregulation of its downstream target, FOSL2 [63]. Similarly, SNHG22 confers resistance to cisplatin and PTX via modulating miR-2467/Gal-1 signaling cascade [64]. Moreover, HOTAIR induces cisplatin resistance by sponging miR-138-5p, thereby halting its binding to EZH2 (enhancer of zeste 2 polycomb repressive complex 2 subunit) and SIRT1 (sirtuin 1) [54]. On the contrary, some lncRNAs may have tumor suppressor effects while their downstream target miRNAs may act as oncogenes. For instance, the overexpression of XIST suppressed tumor proliferation and invasion, and enhanced cisplatin chemosensitivity in CAOV3 and OVCAR3 OC cell lines as well as a xenograft mouse model and this antineoplastic effect was attributed to sponging the oncogenic miR-214-3p [125]. Other lncRNAs such as LINC01125 enhances cisplatin and PTX chemosensitivity through regulating the miR-1972/apoptosis pathway [65].

#### 5.1.8. Other lncRNAs Implicated in Platinum Resistance

Teschendorff et al. reported that OC patients overexpressing HOTAIR, or an equivalent HOTAIR-associated DNA methylation, develop carboplatin resistance and are associated with poor survival outcome [57]. Accordingly, the authors speculated that HOTAIR could be used as both a biomarker and a therapeutic target in carboplatin resistance [57]. Furthermore, it was reported that the plasma level of CASC11 is elevated in ovarian squamous cell carcinoma patients compared to healthy women. Importantly, the plasma level of CASC11 was measured before, 3 and 6 months after initiating chemotherapy (cisplatin, carboplatin, oxaliplatin, and tetraplatin) and the results revealed that the plasma level of CASC11 was significantly elevated after chemotherapy treatment. By applying the UWB1.289 cell line, silencing CASC11 improved the sensitivity of the four agents, suggesting that CASC11 may serve as a potential target to improve therapeutic response [68]. lncRNA BC200, also known as BCYRN1, has been originally identified as a brain-specific cytoplasmic ncRNA that contains 200 nucleotides [126]. Wu et al. explored the role of BC200 in ovarian tissues derived from both 10 normal and 12 OC patients. The results showed that BC200 was downregulated in OC tissues compared to normal samples. Likewise, carboplatin treatment induced BC200 expression and silencing BC200 significantly promoted proliferation and enhanced carboplatin resistance in both SKOV3 and A2780 cell lines [67]. Other lncRNAs, such as SNHG15, were also associated with proliferation, invasion, metastasis, and cisplatin resistance in OC; however, the exact mechanism that accounts for such an effect remains elusive [69].

### 5.2. lncRNAs and Taxanes Resistance

Taxanes exhibit unique pharmacological features as inhibitors of mitosis by binding to the βII subunit of tubulin dimers to promote, rather than suppress, microtubule polymerization. This unique mechanism blunts microtubule disassembly, a necessary event in cell division, ultimately leading to mitotic arrest and apoptosis [127]. PTX and its congener, docetaxel, are considered the mainstay for the treatment of different tumors including head and neck, ovarian, breast, gastrointestinal, and lung cancers [128]. In OC, PTX is used mainly combined with a platinum-based agent as a first-line treatment, or as a single antineoplastic agent in platinum-resistant patients. However, the emergence of resistance is the major limitation for the successful use of taxanes in OC [9]. Thus, understanding the molecular mechanisms encountered in taxanes resistance is critical to improve therapeutic intervention. Notably, the general mechanisms of resistance include increased expression of the MDR-1 gene and its related product P-glycoprotein, the development of βII-tubulin mutations, the upregulation of βIII-isoform of tubulin, and increased production of survivin, an antiapoptotic protein, and α aurora kinase, an enzyme that enhances accomplishment of mitosis [129,130]. Unlike platinum compounds, the studies investigating the role of lncRNAs in the emergence of taxanes resistance in OC are few, probably due to the rare use of taxanes as a single agent in OC. Here, we collected the substantial studies exploring the role of lncRNAs in the development of taxanes resistance. A list of lncRNAs associated with taxane resistance is summarized in Table 2. Besides, the molecular mechanisms implicated in taxanes resistance are illustrated in Figure 5.

#### 5.2.1. Increased Drug Efflux

A study was conducted to explore the association of particular lncRNAs in the emergence of multidrug resistance (PTX, cisplatin, and epirubicin) in both OC and colorectal cancer via using both lncRNA sequencing and bioinformatics analysis [143]. The results identified five upregulated and 21 downregulated lncRNAs as multidrug resistant. Of those lncRNAs, OIP5-AS1 was downregulated in both ovarian and colorectal cancer cell lines, whereas RFPL1S, ZEB1-AS1, and C17orf82 were downregulated in only OC cells with no significant changes in colorectal cancer cell lines, suggesting that MDR-related lncRNAs have both specific and conservative functions among different types of cancers. Additionally, an lncRNA-mRNA co-expression network revealed that lncRNA CTD-2589M5.4 is upregulated in multidrug-resistant cells and co-expressed with multidrug-resistant genes *ABCB1, ABCB4, ABCC3,* and *ABCG2*, advocating its significant role in the development of multidrug resistance [143]. Although this study showed the specific and conservative nature of MDR-linked lncRNAs in ovarian and colorectal cancers, it did not prove whether MDR-linked lncRNAs are the major driver for the emergence of chemoresistance. Thus, further mechanistic studies are needed to explore such an approach. Wang et al. showed that UCA1 confers resistance to PTX in OC cells, and this action could be attributed to the suppression of miR-129, a tumor suppressor miRNA. As a part from its action, miR-129 attenuates tumor growth by silencing the expression of *ABCB1*. Accordingly, UCA1 conveyed resistance to PTX through sponging miR-129, which in turn exacerbated ABCB1-induced drug efflux [131]. Other lncRNAs, such as LINC01118, have been reported to promote PTX resistance via regulating the miR-134/ABCC1 axis [133].

#### 5.2.2. Epithelial–Mesenchymal Transition

NEAT1, a nuclear-restricted lncRNA, positively regulates tumorigenesis in different solid tumors [144]. It was reported that NEAT1 is overexpressed in PTX-resistant OC tissues and cells [134]. Further mechanistic analyses revealed that NEAT1 mediates PTX resistance via upregulating ZEB1 expression by sponging miR-194, a tumor suppressor miRNA. Interestingly, miR-194 knockdown or NEAT1 overexpression markedly abrogated the beneficial effects of ZEB1-knockdown on PTX resistance, indicating that ZEB1 is a potential downstream target of NEAT1 and miR-194 [134]. In conclusion, since ZEB1 is a transcription factor that favors EMT, NEAT1 enhances PTX resistance via activating EMT.

#### 5.2.3. Regulation of Survival Pathways

Activation of survival pathways is a common mechanism by which various lncRNAs can induce chemoresistance. Of those, the ERK pathway was involved in lncRNA-mediated PTX resistance. Generally, the pathway structure includes a small G protein (RAS) and three protein kinases (RAF, MEK, ERK). Activation of this pathway culminates with the translocation of ERK into the nucleus, with a subsequent activation of transcription factors implicated in gene expression and survival [145]. Several reports have shown the relationship between lncRNAs and MAPK pathways in the emergence of PTX resistance. For example, hepatocellular carcinoma upregulated EZH2-associated lncRNA (HEIH), initially identified in hepatocellular carcinoma, promotes tumor growth through suppressing cell differentiation in the G0/G1 phase [146,147]. Guo et al. elucidated that HEIH is upregulated in EC cells and associated with increased PTX resistance [138]. Furthermore, HEIH knockdown dampened tumor proliferation and improved PTX chemosensitivity via blocking the activation of the MAPK signaling pathway [138]. Several lncRNAs have been reported to act as tumor suppressor genes. Of those, lncRNA fer-1-like family member 4 (FER1L4) is downregulated in several cancers, including osteosarcoma [148], esophageal cancer [149], endometrial cancer [150], hepatocellular carcinoma [151], and gastric cancer [152]. Liu et al. showed that FER1L4 expression is reduced in OC cell lines, particularly those resistant to PTX, and overexpression of FER1L4 promotes PTX sensitivity and reduce proliferation [139]. Mechanistically, overexpression of FER1L4 inhibits the phosphorylation MAPK. Additionally, treatment with SB203580, a suppressor of the p38 MAPK pathway, markedly improved PTX sensitivity, suggesting that FER1L4 improves PTX sensitivity via blocking activation of the MAPK pathway [139].

#### 5.2.4. Regulation of Apoptosis

Various lncRNAs regulate the emergence of chemoresistance via modulating apoptosis. For instance, LINC00312 was shown to counteract PTX resistance in OC and enhance apoptosis via blocking BCL-2 and activating the Bax/Caspas3 apoptotic pathway [140]. Another lncRNA, CDKN2B-AS, also called antisense ncRNA in the INK4 locus (ANRIL), is overexpressed in a wide array of solid tumors such as hepatocellular carcinoma [153] and cervical [154], breast [155], endometrial, and prostate cancers [156]. In EC, CDKN2B-AS is upregulated in EC tissues and cells and positively correlated with a high pathological grade of EC and PTX resistance [141]. Likewise, silencing CDKN2B-AS attenuated PTX resistance and enhanced miR-125a-5p expression in EC cells. miR-125a-5p belongs to the miR-125 family and negatively regulates tumorigenesis [157,158]. Mechanistically, CDKN2B-AS induces chemoresistance via regulating the miR-125a-5p/BCL2 and miR-125a-5p/MRP4 network [141].

#### 5.2.5. Regulation of Cell Cycle

Since taxanes are mitotic (M) phase-specific, the regulation of cell cycle plays a fundamental role in taxanes resistance. Zhang et al. demonstrated that lncRNA KB-1471A8.2 is downregulated in OC cells, particularly those resistant to PTX [142]. Moreover, the overexpression of KB-1471A8.2 attenuated cell proliferation, migration, and invasion and enhanced apoptosis and PTX sensitivity in OC cells. Mechanistically, given that PTX exerts its anticancer effect through inhibiting mitosis and inducing apoptosis, KB-1471A8.2 has been shown to improve PTX sensitivity by inhibiting S-phase entry and cyclin-dependent kinase 4 (CDK4) [142]. However, more studies are needed to reveal the full mechanistic pathway encountered in PTX resistance in OC.

#### 5.2.6. Sponging miRNAs

Small nucleolar RNA host gene 5 (SNHG5) is an lncRNA whose gene is located at the chromosomal translocation breakpoint involved in B-cell lymphoma [159]. According to data obtained from TCGA, SNHG5 is downregulated in OC patients, particularly those resistant to PTX, and its downregulation is associated with poor survival and cancer progression [136]. Functional studies revealed that overexpression of SNHG5 dramatically enhanced the PTX sensitivity and suppressed tumor growth in vitro and in vivo. On a mechanistic level, SNHG5 exacerbated PTX sensitivity through acting as a decoy for miR-23a, an oncogenic miRNA overexpressed in multiple tumors [160,161]. Accordingly, this study speculated that the SNHG5/miR-23a axis could be used as a potential target for improving PTX sensitivity in OC [136]. Another study demonstrated that UCA1 enhances proliferation, migration, invasion, and PTX resistance, and this action could be attributed to the regulation of the miR-654-5p/SIK2 axis [132]. Mechanistically, UCA1 acts as a sponge for miR-654-5p, and the knockdown of UCA1 augmented the inhibitory effect of miR-654-5p on the development of OC [162]. Additionally, overexpression of miR-654-5p ameliorated PTX resistance via blunting the expression of salt inducible kinase 2 (SIK2). SIK2 is a member of the adenosine 5′-monophosphate-activated protein kinase (AMPK) subfamily that positively regulates tumor growth and metastasis [163,164]. Remarkably, silencing SIK2 improved PTX sensitivity and inhibited tumor growth, and this action was abolished by miR-654-5p inhibition, suggesting that SIK2 is a downstream target of UCA1/miR-654-5p [132]. Similarly, lncRNA SDHAP1 was reported to promote PTX resistance via regulating the miR-4465/EIF4G2 axis [137]. Moreover, Dong et al. investigated the mechanistic pathway underlying the action of NEAT1 in EC [135]. Functional assays showed that silencing NEAT1 markedly mitigated proliferation, invasion, and sphere formation and improved PTX sensitivity in EC. Mechanistically, NEAT1 induces its action via regulating the miR-361/STAT3 axis. Importantly, miR-361 attenuates proliferation, invasion, and PTX resistance by silencing signal transducer and activator of transcription 3 (STAT3) [135]. STAT3 is a transcription factor that positively regulates carcinogenesis [135]. Moreover, NEAT1-induced inhibition of miR-361 activates not only the STAT3 pathway but also several prometastatic and tumor microenvironment–related genes such as *MEF2D, ROCK1, WNT7A,* and *KPNA4*. These genes play a fundamental role in the acquisition of metastatic potential and PTX resistance in EC cells. Therefore, targeting NEAT1 could be a promising approach for improving therapeutic response to PTX [135].

#### 5.2.7. Other lncRNAs Implicated in Paclitaxel Resistance

A comprehensive study has been conducted to reveal the expression profile of lncRNAs associated with PTX resistance in OC tissues and cell lines. The findings of this study identified seven lncRNAs aberrantly expressed in PTX-resistant phenotypes: XR_938392, XR_947831, XR_948297, XR_938728, NR_073113, NR_103801, and NR_036503. Furthermore, functional analysis revealed that the signature for these lncRNAs is positively correlated with a bundle of 129 genes implicated in insulin secretion-related pathway. Consequently, the authors speculated that those lncRNAs regulate tumor proliferation and chemoresistance via an interplay between cancer-related pathways and insulin secretion and that these lncRNAs could be used as predictive biomarkers for PTX resistance in OC patients [165]. However, more studies need to be conducted to validate the use of these lncRNAs as predictive biomarkers for PTX resistance in OC and to explore the underlying mechanistic circuitries of the correlation between these lncRNAs and chemoresistance. Another study has identified a strong correlation between XIST RNA expression and PTX sensitivity in OC, suggesting that XIST could be used as a prognostic biomarker for chemosensitivity in OC [166]. However, this study lacked the potential mechanism for such a correlation, which warrants further investigation.

## 6. Therapeutic Approaches for Targeting lncRNAs

Several lncRNAs significantly modulate tumorigenesis and chemoresistance and could be used as therapeutic targets in clinical settings. For a successful translation into the clinic, novel effective approaches for targeting lncRNAs should be established. Currently, multiple strategies can be used to effectively target lncRNAs, including (1) the post-transcriptional degradation of lncRNAs, (2) the modification of lncRNA genes, (3) and the loss of lncRNAs’ function [167].

Post-transcriptional targeting of lncRNAs relies on nucleic acid–based modalities that have the ability to target any unique portion of the transcriptome, thereby affecting the “undruggable” parts of the entire genome. Presently, there are two major strategies harnessing nucleic acid-based techniques: double-stranded RNA-mediated interference (RNAi) and single-stranded antisense oligonucleotides (ASOs). RNAi comprises application of siRNA, a short dsRAN molecule, approximately 21–23 nucleotides in length, that interacts with and activates the RNA-induced silencing complex (RISC). Importantly, the endonuclease component of the RISC, named Argonaute 2 (Ago2), cleaves the sense strand of siRNA, enabling the antisense strand to bind to mRNA in a complementary manner, leading to mRNA degradation [168]. Since double-stranded RNAs are inherently vulnerable to degradation by nucleases, chemical modifications are required to protect them. Among these modifications, the addition of 2′-O methyl (2′-O-Me) sugar residues and phosphorothioate linkages on the 3′ end of the RNA markedly improves the pharmacological features of siRNA-based approaches [169]. Although numerous in vitro studies have been conducted via using RNAi as a tool for targeting lncRNAs, the use of siRNA/shRNAs in preclinical studies is challengeable due to poor bioavailability and ineffective delivery methods [167]. Unlike RNAi, ASO is a single-chain oligonucleotide that binds to RNA through Watson-Crick base pairing, resulting in activation of an RNase H–dependent cleavage mechanism, eventually leading to endo-nucleolytic cleavage of RNA [167]. The advances in ASO chemistry have led to a great impact on improving the overall pharmacological properties of ASOs via protecting them from cleavage by nucleases, enhancing binding affinity, and optimizing pharmacokinetic features [170,171]. Currently, ASOs have been involved in several clinical trials, and they are emerging as a promising therapeutic strategy for targeting lncRNAs [172].

The evolution of technology has led to the development of a highly advanced, versatile method for genome editing, known as the clustered regularly interspaced short palindromic repeats (CRISPR)-Cas system. This system consists of two fundamental parts: an enzyme called Cas and a predesigned RNA sequence named guide RNA. Cas has the ability to induce site-specific cleavage of DNA, while a guide RNA binds to both DNA strands in a complementary manner and directs the Cas enzyme to the right region of the genome [173]. By applying such technology into lncRNAs, it is feasible to induce silencing of lncRNA-expressing loci, thereby blocking lncRNA transcription. On a mechanistic level, dead-Cas9 is merged with transcriptional repressors and then directed into a specific gene promoter to block lncRNA transcription [174,175]. Furthermore, recent studies have identified CRISPR-Cas13 as another promising tool to block lncRNAs transcription [176]. Although the CRISPR-Cas approach is included in many preclinical studies, translation of this approach into the clinical setting warrants further investigation [167].

Emerging evidence reveals that certain lncRNAs exert their functions by forming secondary/tertiary structures or by binding to proteins. For instance, the 3′ ends of NEAT1 and MALAT1 form tertiary structures by folding into a unique triple helical structure [167,177]. In addition, numerous nuclear lncRNAs bind to chromatin-modifying complexes and regulate chromatin remodeling. Given these lncRNA functions, a new strategy has been evolved to prevent either lncRNA-protein interaction or formation of lncRNA secondary/tertiary structures. In this setting, specific types of ASOs that cannot stimulate RNAse H activity can be used to bind specifically to lncRNA, consequently blocking lncRNA-protein interactions [167]. Moreover, small molecule inhibitors can be used to target unique structural components of lncRNAs required for forming secondary/tertiary structures, thereby destabilizing the transcript and blocking their functions; however, this feature needs further investigation [167].

Natural antisense transcripts (NATs) are a class of ncRNAs transcribed in an antisense direction of a protein-coding gene [178]. Importantly, these transcripts negatively regulate the expression of neighboring protein-coding genes via acting in cis. Hence, targeting NATs using ASOs results in the upregulation of the neighboring protein-coding genes [179]. Accordingly, silencing NATs neighboring tumor suppressor genes could lead to the upregulation of tumor suppressor molecules and therefore negatively regulate tumorigenesis [167]. For instance, ANRIL and P21-AS lncRNAs are located near critical tumor suppressor genes *CDKN2B* and *CDKN1A*, respectively, and knockdown of these NATs could be a potential therapy to inhibit tumor growth in multiple malignancies [167,180,181].

## 7. lncRNAs as Diagnostic and Prognostic Biomarkers

The discovery that several lncRNAs are over-expressed in different cancers and can predict poor prognosis has drawn an increasing attention given their potential to be used as a clinical tool in diagnosis and/or prognosis of OC. Multiple approaches such as microarray hybridization, quantitative reverse transcription polymerase chain reaction (qRT-PCR), FISH and genome-wide profiling have been utilized to determine the levels of lncRNAs in different biological samples and several lncRNAs have been functionally characterized [182]. Since they have a cancer-specific expression pattern, lncRNAs have a great potential not only as novel therapeutic targets, but also as diagnostic and/or prognostic biomarkers [183,184]. Our group recently reported that multiple genomic non-coding loci harboring human-specific/primate-specific motifs, named pyknons, are differentially transcribed between healthy and diseased tissues in chronic lymphocytic leukemia and colorectal cancer. Furthermore, multiple new loci whose expression correlates with the overall survival of colorectal cancer patients’ have been identified, suggesting that pyknons could be used as a primate-specific biomarker [183,184]. Similarly, the identification of primate-specific lncRNAs could be used as diagnostic/prognostic biomarker for multiple cancers. Various reports indicated that lncRNAs are aberrantly expressed in OC patients and such dysregulated expression correlates with clinicopathological parameters, including metastasis, histological type, and tumor size [185]. In the context of histology, several lncRNAs are correlated with different epithelial subtypes, and therefore they can be used to distinguish between different types of OC [185]. For instance, Casc2 and FLJ3360, respectively, differentiate the serous and high-grade serous from the other subtypes [186,187]. Furthermore, SNHG15 was found to be highly expressed in patients with type II cancers than patients with type I cancers indicating that it could distinguish between type I and type II OC [69].

In addition to their diagnostic value, certain lncRNAs can be used to predict survival of OC patients based on differential expression levels in specific cohort studies, resulting in different survival outcomes between patients with high vs. low expression level of lncRNAs [185]. Recently, a meta-analysis has indicated that aberrant expression of lncRNAs is positively correlated with poor survival outcomes in OC. During this study, 11 lncRNAs, including NEAT1, HOTAIR, MALAT1, TC010441, ANRIL, AB073614, CCAT2, ZFAS1, UCA1, SPRY4-IT1, and HOXA11as, were significantly associated with reduced overall survival, progression-free survival, and disease-free survival in OC patients [188]. Similarly, Luo et al. revealed that upregulation of eight lncRNAs; HOTAIR, NEAT1, CCAT2, UCA1, C17orf91, ANRIL, TC0101441, and AB073614 and downregulation of 5 lncRNAs; ASAP1-IT1, AC104699.1.1, LINC00472, FAM215, and RP11-284N8.3.1 were correlated with poor prognosis in OC given a probability for harnessing these lncRNAs as potential prognostic biomarkers for OC [189]. An integrated competing endogenous RNA network analysis has been conducted via implementing RNA-sequencing of data retrieved from the TCGA database and GSE17260 dataset to identify the potential RNA signatures that can be used as prognostic biomarkers for recurrent OC. Three lncRNAs, WT1-AS, NBR2, and ZNF883, were highly correlated with recurrent OC, indicating that these lncRNAs could be used as prognostic biomarkers for recurrent OC [190].

As mentioned above, platinum- and taxane-based chemotherapy is still the cornerstone for the treatment of OC patients. However, the overall outcome is still unsatisfactory owing to the development of chemoresistance. Given the fact that lncRNAs play a substantial role in the emergence of acquired resistance to different chemotherapy, several lncRNAs can be utilized as a clinical tool for stratifying patients into responders and non-responders and therefore optimizing the overall response to chemotherapy. For instance, SNHG5 can be used as a predictor for paclitaxel resistance, while RP11-135L22.1 and LINC00515 are very effective regarding platinum resistance [69].

## 8. Challenges for lncRNAs Applications in Cancer

Although a growing body of evidence indicates the significant role of certain lncRNAs in cancer, we still face multiple challenges that may hamper the impact of lncRNAs in the field of cancer. Some of these challenges are linked to the intrinsic features of lncRNAs, whereas others are associated with the approaches harnessed to either accurately measure the levels of lncRNAs or effectively target lncRNAs [188]. One of the major challenges is that lncRNAs are expressed in lower level compared to protein-coding genes [189]. In fact, this low expression could be partially attributed to the specific pattern of lncRNAs expression in different cells and tissues. Although some scholars believe such a specific expression pattern is an advantageous, it seems to be one of the major challenges, at least experimentally, since low abundant genes need more sophisticated approaches for accurate quantification [190]. Thus, a larger sample size and/or advanced technology is required to accurately measure the expression level of lncRNAs, consequently minimizing the errors and optimizing the overall analysis process [191]. Interestingly, advances in research revealed an attractive technology, named lnc(RNA) capture sequencing, that has the ability to accurately detect the expression level of lncRNAs in different biological samples with high sensitivity and reproducibility [192]. This technology utilizes a predesigned biotinylated probe that specifically capture the (lnc)RNA of interest while depleting more abundant mRNAs from the sequencing library [193]. We think the development of such technologies, despite being expensive, may tackle this challenge enabling for accurate detection of lncRNAs in different biological samples.

As mentioned earlier, lncRNAs have the potential to be employed as a diagnostic or prognostic biomarker. Importantly, to be an effective biomarker, lncRNAs should be detected in different body fluids particularly plasma, urine, and cerebrospinal fluid. However, a major challenge has been raised toward lncRNAs that the majority of lncRNAs detected in plasma are partially or completely fragmented [194]. This fact leads the scientists to pay much attention for exploring alternative effective approaches for the detection of lncRNAs in different biological fluids [195]. One of these approaches is the detection of lncRNAs enclosed in extracellular vesicles, particularly exosomes [195,196]. Exosomes are cell-derived extracellular vesicles, surrounded by lipid bilayers, and commonly found in plasma, urine, and cerebrospinal fluid [195]. They represent an attractive source of diagnostic/prognostic biomarkers since they harbor a large variety of biomolecules, e.g., DNA and lncRNAs reflecting a real-time status for the original cells. Likewise, they can be retrieved in non-invasive manner, can protect lncRNAs from degradation by ribonuclease activity, have a higher concentration compared to circulating cancerous cells and nucleic acids, and can easily penetrate into different body fluids due to their small size [195]. Several studies highlighted the role of exosomal lncRNAs as a biomarker in cancer. For instance, Zaho et al. reported that the overexpression of lncRNA HOTTIP is associated with poor survival in gastric cancer patients and serum exosomal HOTTIP is more accurate, as a diagnostic biomarker, than the commonly used tumor biomarkers carcino-embryonic antigen and carbohydrate antigens19-9 and 72-4 [197]. Although exosomes have several features that render them as plausible diagnostic/prognostic biomarkers, the translation of exosomal lncRNAs as diagnostic biomarkers into the clinical setting remains challenging. The major challenges for exsosomal application are exosomal heterogeneity, discrepancies between in vivo and in vitro experiments, difficulty to determine their origin, and a lack of tumor specificity [195,198]. Whether exosomal or plasma lncRNAs can serve as diagnostic biomarkers warrants further investigations.

Another challenge that hinders the application of lncRNAs is that many lncRNAs are dispensable and lack a definitive function. Recently, a study has been conducted in zebrafish by selecting 25 lncRNAs based on specific features such as expression profiling, conservative expression or vicinity to developmental regulators [199]. These lncRNAs were further subjected to CRISPR-Cas9 to induce 32 deletion alleles. Interestingly, the extrapolation from the results assumed that the majority of zebrafish lncRNAs have no overt roles in terms of fertility, embryogenesis, and viability [199]. Whether such finding affects the application of lncRNAs in cancer, in general, and chemoresistance, in particular, warrants further investigations. However, the expression of lncRNAs has been shown to be species specific [200]. An across-mammalian-genomes analysis demonstrated that approximately 30% of lncRNA transcripts (*n* = 4546) are primate specific (human, macaque, chimp, marmoset, and orangutan) [201]. Another study has been conducted via characterizing human lncRNA expression patterns in nine tissues across six mammalian species and multiple individuals [202]. Interestingly, the extrapolation from the results showed that of the 1898 human lncRNAs expressed in nine tissues, orthologous transcripts for 80% in chimpanzee, 63% in rhesus, 39% in cow, 38% in mouse, and 35% in rat have been identified [202]. Recently, two primate-specific lncRNAs, N-BLR and FLANC, have been shown to be conserved in human and to play a significant role in the development of colorectal cancer [203,204]. Overall, we conjecture that such discrepancy in lncRNAs function could be attributed to poor conservation of lncRNAs among species and that the closer the species to human, the more conservative transcripts of lncRNAs. Larger comparative studies are highly recommended to further reveal such discrepancy.

Some limitations that may prevent the translation of lncRNAs into the clinic are related to lncRNA-targeting approaches. As elaborated earlier, the most common approaches for targeting lncRNAs are RNA-based therapy and a CRISPR Cas system. Although RNA-based drugs can be easily applied into the cells, their systemic administration is vulnerable to degradation by nucleases and can provoke immune response. Moreover, they cannot penetrate the cell membrane easily due to negatively charged structure and can produce off-target effects [205]. Nonetheless, a wide variety of delivery agents are in development to ensure the efficient uptake of oligonucleotides, and consequently minimizing systemic toxicity. Our group has developed a neutral nanoliposome, named 1,2-Dioleoyl-sn-Glycero-3-Phosphatidylcholine (DOPC), to deliver nucleic acid into target cells. Intriguingly, our results revealed that DOPC nanoliposome has a 10-fold higher efficiency in delivering nucleic acid compared to cationic liposomes [206]. Furthermore, our group are currently undergoing a phase I trial that studies the effect of EphA2 siRNA/DOPC liposomal formulation on treating patients with advanced or recurrent solid tumors. Another lncRNA-based therapeutic strategy that has the potential for clinical application is the use of a BC-819 plasmid/polyethylenimine (PEI) system. BC-819, also known as H19-DTA, is a DNA vector that encodes for Diphtheria Toxin A fragment (DTA) under the control of H19 gene promoter (BC-819 or DTA-H19), whereas PEI is a transfection agent. This strategy is based on the fact that H19 lncRNA is overexpressed in malignant cells with minimal expression in normal cells, and, therefore, only malignant cells have the ability to activate H19 gene promoter and DTA production. Eventually, the resultant DTA disrupts protein synthesis and selectively destroys tumor cells. Clinical trials revealed that BC-819 administered locally in combination with systemic chemotherapy could improve therapeutic outcomes for the treatment of pancreatic, ovarian, or peritoneal cancer [207,208,209].

## 9. Conclusions and Future Perspectives

Dysregulation of lncRNAs can substantially trigger or ameliorate the emergence of resistance to platinum- and taxane-based chemotherapy in OC. On a mechanistic level, lncRNAs can modulate the response to chemotherapy via regulating drug uptake and efflux, intracellular drug detoxification, DNA damage repair, autophagy, apoptosis, EMT, cell cycle progression, and survival pathways. These findings can pave the path toward the use of an lncRNA-based approach for enhancing therapeutic response to platinum- and taxane-based chemotherapy in OC.

The use of lncRNA-based approach in OC is classified into diagnostic and therapeutic approaches. As diagnostic or prognostic tools, several lncRNAs can be implemented as prognostic biomarkers for both carcinogenesis and chemoresistance, given the facts that most lncRNAs have differential and specific expression, are stable in most body fluids like serum, urine, and saliva, and can be easily detected by experimental techniques such as microarray hybridization, qRT-PCR, and RNA sequencing. Although several lncRNAs are subjected to degradation by ribonucleases in plasma, exosomal lncRNAs are stable and could be used as diagnostic or prognostic biomarkers. However, the application of exosomes has several limitations such as tumor heterogeneity, lack of tumor specificity, and difficult isolation. Whether exosomal or plasma lncRNAs can serve as diagnostic biomarkers warrants further investigations. Advances in technology may provide a new technique for easily isolating and purifying exosomal lncRNAs that can be further utilized as unique biomarkers for individual cancers. Several characteristics are required for lncRNAs to be used as successful diagnostic and/or prognostic biomarkers. First, lncRNAs should exhibit a unique, specific signature for individual cancer types. Furthermore, they should occur in an early stage of cancer and show statistically significant expression compared to normal patients. In addition, they should be expressed in significant amounts that can be easily detected by quantitative methods. Finally, they should have standard normalization methods along with reliable internal controls.

The therapeutic approach is less advanced. Although the efficacy of certain lncRNAs as potential targets for ameliorating chemoresistance has been verified experimentally, the effective translation of these lncRNAs into clinical practice and the approval of lncRNA-targeting strategy have not been achieved yet. However, the use of DOPC nanoliposomal formulation and other neutral nanoliposomes could enhance the uptake of oligonucleotides into cancer cells, eventually leading to improved selectivity while minimizing systemic toxicity. Interestingly, our laboratory in collaboration with others are undergoing a clinical trial via harnessing DOPC nanoliposomal siRNA formulation for targeting EphA2 in OC and other solid tumors. This formulation could pave the path toward establishing a similar technique to target lncRNAs for therapeutic purpose. Moreover, the use of BC-819/PEI shows promising results when combined with H19 in pancreatic and OC clinical trials. Overall, we think that the development of an RNA-based drug to target particular lncRNA is plausible based on the previous research along with advances in technology.

The majority of studies investigating the role of lncRNAs in the emergence of chemoresistance encounter several limitations. First, these studies lack the detailed molecular mechanisms underlying the development of resistance. Second, most of these studies have been conducted using in vitro models only, while animal models or patient samples have not been investigated. Third, the translation of lncRNA-based approaches to circumvent chemoresistance will be hindered by poor sequence conservation among different species. In this setting, the lncRNAs that have been verified in mouse models may not be successfully translated into clinical application. Thus, considering these limitations, subsequent research should be directed toward the identification of additional lncRNAs along with investigating their roles in carcinogenesis and chemoresistance.

Despite the challenges facing lncRNAs, several approaches should be taken into consideration for optimizing lncRNA impacts on cancer. These approaches, from our perspective, include: (1) performing large projects for annotation and functional characterization of more lncRNAs in human; (2) detection of lncRNAs that show significantly aberrant expression in cancer vs. normal samples; (3) generating and unifying robust bioinformatic data that can be used as a tool for analyzing lncRNAs; (4) focusing on the highly conserved lncRNAs for further mechanistic studies; (5) testing the new delivering formulations that enhance selective uptake of RNA-based drugs in a large number of in vivo experiments; and (6) selecting the most promising lncRNAs for undergoing clinical trials in cancer patients.

Overall, accumulating evidence indicates that several lncRNAs are implicated in carcinogenesis and chemoresistance, and some of these lncRNAs can be implemented as either diagnostic biomarkers or therapeutic targets for chemoresistance in OC. Advances in technology may allow for effective translation of lncRNA-based therapeutic approaches into clinical practice.

## Figures and Tables

**Figure 1 cancers-12-02406-f001:**
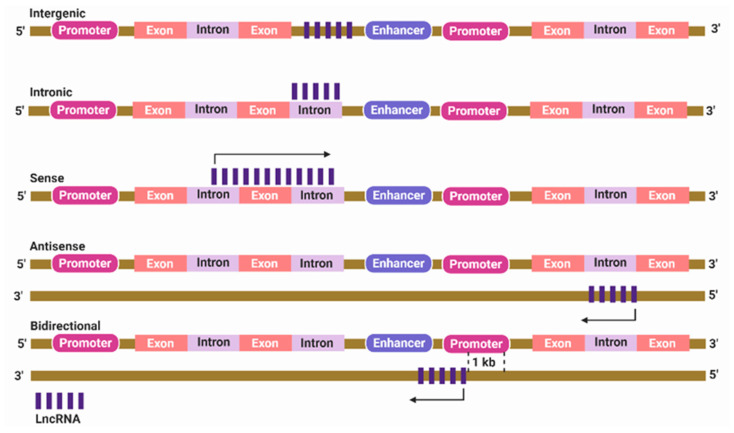
Classification of long non-coding RNAs (lncRNAs) based on their genomic location. Intergenic lncRNA is located between two protein-coding genes. Intronic lncRNA lies entirely within the intron of a protein-coding gene. Sense lncRNA is transcribed from the same strand and in the same direction as the nearby protein-coding genes. Antisense lncRNA is transcribed from an opposite strand. Bidirectional lncRNA is located within 1 kb of the promoter region of a protein-coding gene but is transcribed from an opposite strand.

**Figure 2 cancers-12-02406-f002:**
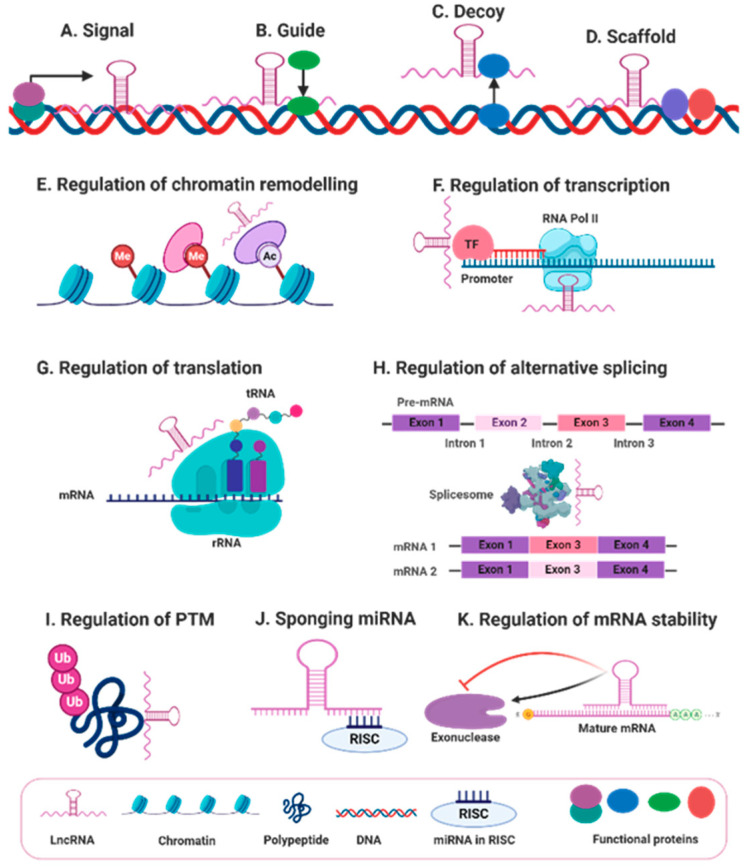
General biological functions of lncRNAs. lncRNA biological functions are generally categorized into four main archetypes of molecular mechanisms, including (**A**) signals, (**B**) guides, (**C**) decoys, and (**D**) scaffolds. In addition, lncRNAs exert diverse regulatory functions originating from these archetypes such as regulation of; (**E**) chromatin remodeling via modulating recruitment of epigenetic factors, (**F**) transcription through controlling transcription factors or RNA polymerase 2 (RNA pol II), (**G**) protein translation, (**H**) alternative splicing and other post-transcriptional modifications, (**I**) protein ubiquitination and other post-translational modifications (PTMs), (**J**) miRNA-induced gene silencing via sponging miRNA as a part of the RNA-induced silencing complex (RISC), and (**K**) mRNA stability by modulating mRNA-degrading enzymes.

**Figure 3 cancers-12-02406-f003:**
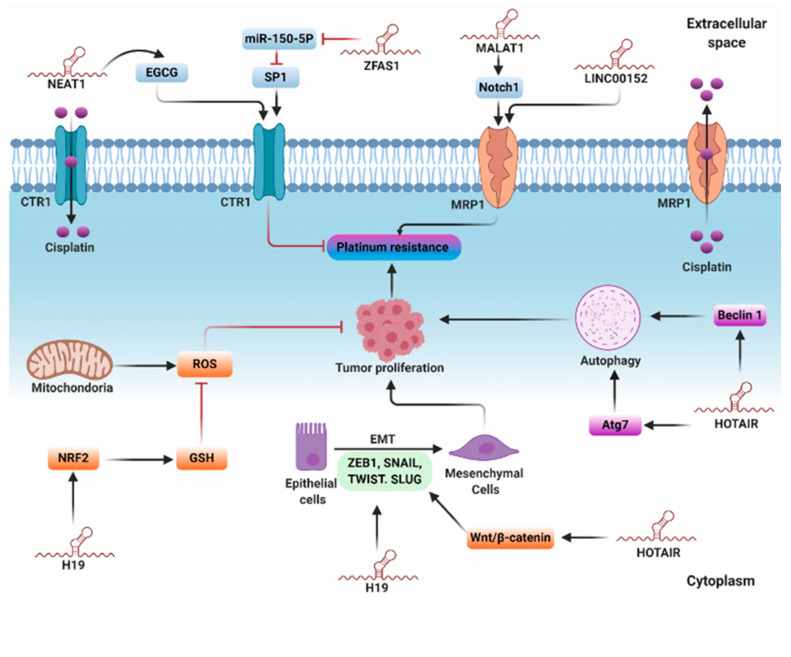
Molecular mechanisms of lncRNAs involved in platinum resistance. The lncRNA ZFAS1 enhances cisplatin resistance through regulating the miR-150-5p/SP1/copper transporter 1 (CTR1) axis. Nuclear-enriched abundant transcript 1 (NEAT1) ameliorates platinum resistance via upregulating epigallocatechin-3-gallate (EGCG)-induced CTR1 expression. Metastasis-associated lung adenocarcinoma transcript 1 (MALAT1) and LINC00152 promote cisplatin resistance through activating MRP1. Homeobox transcript antisense RNA (HOTAIR) enhances platinum resistance via activating autophagy and Wnt/β catenin-induced epithelial–mesenchymal transition (EMT). The lncRNA H19 induces platinum resistance via enhancing the expression of nuclear factor erythroid 2 (NRF2) and glutathione (GSH), thereby preventing reactive oxygen species (ROS)-induced cytotoxicity. Moreover, H19 promotes platinum resistance by activating transcription factors involved in EMT.

**Figure 4 cancers-12-02406-f004:**
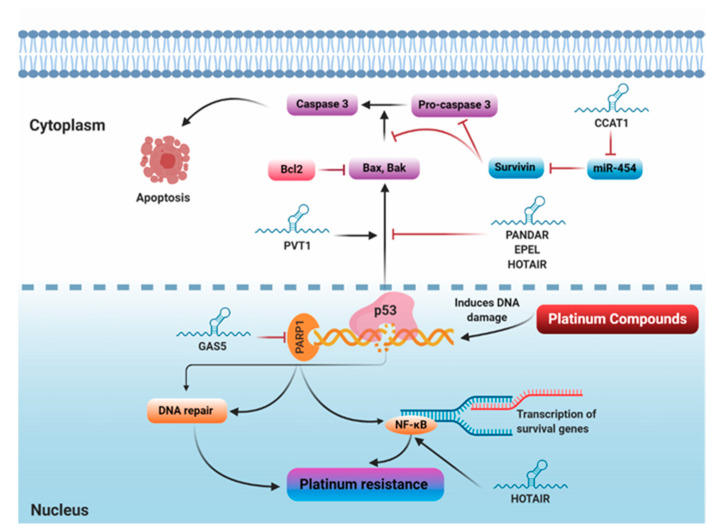
lncRNAs regulate platinum resistance via modulating DNA-damage-induced apoptosis. Growth arrest-specific transcript 5 (GAS5) mitigates cisplatin resistance via inhibiting PARP1-induced DNA repair. Colon cancer-associated transcript 1 (CCAT1) promotes platinum resistance through sponging miR-454, thereby activating survivin and inhibiting apoptosis. HOTAIR exacerbates platinum resistance through activating NF-κB, leading to increased expression of survival genes. lncRNAs promoter of CDKN1A antisense DNA damage activated RNA (PANDAR), E2F-mediated proliferation enhancing lncRNA (EPEL), and HOTAIR induce platinum resistance via inhibiting p53-induced apoptosis, while plasmacytoma variant translocation 1 (PVT1) inhibits platinum resistance via promoting p53-induced apoptosis.

**Figure 5 cancers-12-02406-f005:**
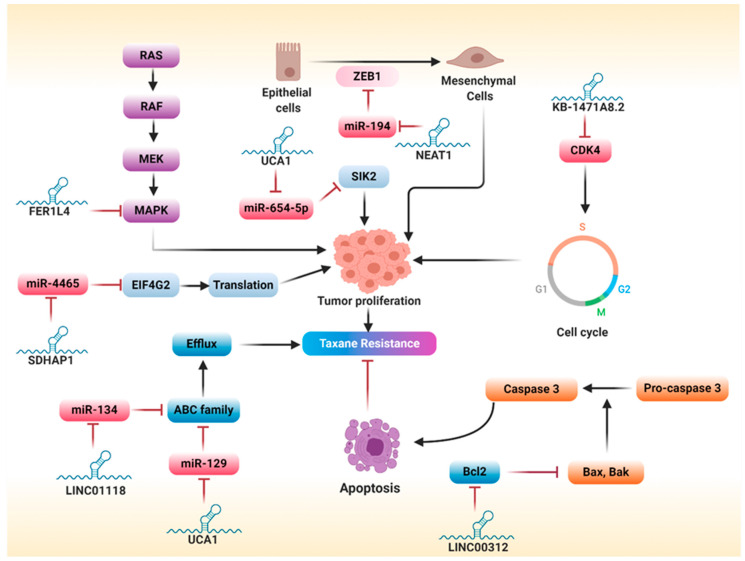
Major lncRNAs involved in taxane resistance. LINC01118 and urothelial carcinoma associated 1 (UCA1) promote taxane resistance via activating ATP-binding cassette (ABC) family and drug efflux. Linc00312 circumvents taxane resistance through suppressing Bcl2 and inducing apoptosis. Succinate dehydrogenase complex, subunit A, flavoprotein pseudogene 1 (SDHAP1) promotes taxane resistance via sponging miR-4465, consequently enhancing EIF4G2 and initiating protein translation. FER1L4 mitigates taxane resistance by suppressing the mitogen-activated protein kinase (MAPK) pathway. UCA1 promotes taxane resistance by regulating the miR-654-5p/SIK2 axis. NEAT1 promotes taxane resistance via sponging miR-194, thereby activating ZEB1 and EMT. KB-1471A8.2 ameliorates taxane resistance by blocking CDK4 and S-phase entry.

**Table 1 cancers-12-02406-t001:** List of lncRNAs associated with platinum resistance.

Name of lncRNA	Classification	Type of Cancer	Role in Platinum Resistance	Mechanism	Reference
ZFAS1 (Zinc finger antisense 1)	Antisense	Ovarian cancer	↑ Cisplatin resistance	Regulating CTR1 expression via regulating miR-150-5p/SP1 axis	[46]
MALAT1 (metastasis-associated lung adenocarcinoma transcript 1)	Intergenic	Ovarian cancer	↑ Cisplatin resistance	Regulating Notch1/ABCC1/MRP1 signaling pathway	[47]
GAS5 (growth arrest-specific transcript 5)	Antisense	Ovarian cancer	↓ Cisplatin resistance	↓ PARP1 and ↓ phosphorylation of ERK, JNK, and MAPK	[48]
NEAT1 (nuclear-enriched abundant transcript 1)	Intergenic	Lung cancer	↓ Cisplatin resistance	Upregulating EGCG-induced CTR1 expression	[49]
H19 (imprinted maternally expressed transcript)	Intergenic	Ovarian cancer	↑ Cisplatin resistance	↑ Expression of NRF2-targeted proteins and ↑ GSH activity → ↓ cisplatin-induced oxidative cytotoxicity	[50]
H19	Intergenic	Ovarian cancer	↑ Cisplatin resistance	↑ TWIST, SLUG, and SNAIL → ↑ EMT	[51]
HOTAIR (HOX transcript antisense RNA)	Antisense	Ovarian cancer	↑ Cisplatin resistance	Activating ATG7 and enhancing autophagy	[52]
		Endometrial cancer	↑ Cisplatin resistance	Activating beclin-1 and enhancing autophagy	[53]
		Ovarian cancer	↑ Cisplatin resistance	Regulating miR-138-5p/EZH2/SIRT1 axis	[54]
		Ovarian cancer	↑ Cisplatin resistance	Activating Wnt/β-catenin pathway → ↑ EMT	[55]
		Ovarian cancer	↑ Cisplatin resistance	HOTAIR ↑ NF-κB, IL-6, and CHK1-p53-p21 → repair of damaged DNA	[56]
		Ovarian cancer	↑ Carboplatin resistance	Unknown	[57]
PANDAR (promoter of CDKN1A antisense DNA damage activated RNA)	Antisense	Ovarian cancer	↑ Cisplatin resistance	Modulating PANDAR/SRFS2/p53 axis → ↓ p53-induced apoptosis	[58]
EPEL (E2F-mediated proliferation enhancing lncRNA)	Intergenic	Endometroid cancer	↑ Carboplatin resistance	↓ Expression of p53 → ↓ apoptosis.	[59]
PVT1 (plasmacytoma variant translocation 1)	Intergenic	Ovarian cancer	↓ Carboplatin and docetaxel resistance	↑ p53 and TIMP1 → ↑ apoptosis and ↓ tumor invasion, respectively.	[60]
CCAT1 (colon cancer-associated transcript 1)	Antisense	Ovarian cancer	↑ Cisplatin resistance	Modulating miR-454/survivin pathway	[61]
LINC00152 (long intergenic non-coding RNA 152)	Intergenic	Ovarian cancer	↑ Cisplatin resistance	↑ MDR1 and MRP1 and ↓ apoptosis	[62]
UCA1 (urothelial carcinoma associated 1)	Intergenic	Ovarian cancer	↑ Cisplatin resistance	Regulating miR-143/FOSL2 axis	[63]
SNHG22 (small nucleolar RNA host gene 22)	Antisense	Ovarian cancer	↑ Cisplatin and paclitaxel resistance	Regulating miR-2467/Gal-1 signaling cascade	[64]
LINC01125	Sense-overlapping	Ovarian cancer	↓ Cisplatin and paclitaxel resistance	Regulating miR-1972/apoptosis pathway	[65]
ENST00000457645	Intergenic	Ovarian cancer	↓ Cisplatin resistance	Unknown	[66]
BCYRN1 (brain cytoplasmic RNA1)	Intergenic	Ovarian cancer	↓ Carboplatin resistance	Unknown	[67]
CASC11 (cancer susceptibility 11)	Intergenic	Ovarian cancer	↑ Cisplatin and carboplatin resistance	Unknown	[68]
SNHG15 (small nucleolar RNA host gene 15)	Intergenic	Ovarian cancer	↑ Cisplatin resistance	Unknown	[69]
CRNDE (colorectal neoplasia differentially expressed)	Intergenic	Ovarian cancer	Upregulated in cisplatin resistance	Unknown	[70]
RP11-1A16.1	Intergenic	Ovarian cancer	Downregulated in cisplatin resistance	Unknown	[70]
AC000035.3	Intergenic	Ovarian cancer	Downregulated in cisplatin resistance	Unknown	[70]
AC003986.7	Antisense	Ovarian cancer	Upregulated in cisplatin resistance	Unknown	[70]
RP11-6N17.4	Bidirectional	Ovarian cancer	Upregulated in cisplatin resistance	Unknown	[70]
PLAC2	Intronic	Ovarian cancer	Upregulated in cisplatin resistance	Unknown	[70]
CTD-2026G22.1	Intergenic	Ovarian cancer	Downregulated in cisplatin resistance	Unknown	[70]
RP11-1A16.1	Intergenic	Ovarian cancer	Downregulated in cisplatin resistance	Unknown	[70]

The arrows in this table refer to: ↑ (increase); ↓ (decrease); → (leading to).

**Table 2 cancers-12-02406-t002:** List of lncRNAs associated with taxane resistance.

Name of lncRNA	Classification	Type of Cancer	Role in Drug Resistance	Mechanism	Reference
UCA1 (urothelial carcinoma associated 1)	Intergenic	Ovarian cancer	↑ PTX resistance	Sponging miR-129 → ↑ ABCB1-induced drug efflux Regulating miR-654-5p/SIK2 axis	[131] [132]
LINC01118	Intergenic	Ovarian cancer	↑ PTX resistance	Regulating miR-134/ABCC1 axis	[133]
NEAT1 (nuclear paraspeckle assembly transcript 1)	Intergenic	Ovarian cancer	↑ PTX resistance	Sponging miR-194 → ↑ ZEB1 expression → ↑ EMT	[134]
NEAT1 (nuclear paraspeckle assembly transcript 1)	Intergenic	Endometrial cancer	↑ PTX resistance	Sponging miR-194 → ↑ STAT3 expression	[135]
SNHG5 (small nucleolar RNA host gene 5)	Intergenic	Ovarian cancer	↓ PTX resistance	Sponging oncogenic miR-23a	[136]
SDHAP1 (succinate dehydrogenase complex, subunit A, flavoprotein pseudogene 1)	Pseudogene	Ovarian cancer	↑ PTX resistance	Regulating miR-4465/EIF4G2 axis	[137]
HEIH (highly expressed in hepatocellular carcinoma)	Intergenic	Endometrial cancer	↑ PTX resistance	Activating MAPK pathway	[138]
FER1L4 (fer-1 like family member 4)	Pseudogene	Ovarian cancer	↓ PTX resistance	↓ Phosphorylation of MAPK	[139]
LINC00312 (long intergenic non-protein coding RNA 312)	Intergenic	Ovarian cancer	↓ PTX resistance	Blocking BCL-2 expression and activating Bax/Caspas3 apoptotic pathways	[140].
ANRIL (antisense non-coding RNA in the INK4 locus)	Antisense	Endometrial cancer	↑ PTX resistance	Regulating both miR-125a-5p/Bcl2 and miR-125a-5p/MRP4 axes	[141]
KB-1471A8.2	Antisense	Ovarian cancer	↓ PTX resistance	Blocking S-phase entry and CDK4	[142]
CTD-2589M5.4	Intergenic	Ovarian cancer	Associated with multidrug resistance	Unknown but may be related to ABCB1, ABCB4, ABCC3, and ABCG2	[143]

The arrows in this table refer to: ↑ (increase); ↓ (decrease); → (leading to).

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
