# Peer review of "Back to the Future: Rethinking the Great Potential of lncRNAS for Optimizing Chemotherapeutic Response in Ovarian Cancer"

_cancers, 2020, doi:10.3390/cancers12092406_

Round 1

Reviewer 1 Report

This review by Elsayed  et al focuses on the rethinking the great potential of lncRNAS for optimizing chemotherapeutic response in ovarian cancer. Be specific, the authors in this review summarizes robust data concerning the involvement of lncRNAs in the emergence of acquired resistance to platinum- and taxane-based chemotherapy in OC. Overall, this review is comprehensive and meaningful.

Author Response

  • Response:

We would like to thank this reviewer so much for his positive comment regarding our manuscript.

Reviewer 2 Report

 Recently, lncRNAs in ovarian cancer have received a great deal of attention and a number of review articles have been reported. They include a paper published in Cancers (PMID; 32326249) and a paper written by the author of this article (PMID; 31847695).

 This review paper itself appears to be well written. The main problem, however, is that it is not clear what is new in the review article compared to many other recently published review articles, including their own.

 Therefore, unfortunately, this review paper does not seem to be worthy of publication in Cancers.

Author Response

  • Reviewer #2 comment:
    “Recently, lncRNAs in ovarian cancer have received a great deal of attention and a number of review articles have been reported. They include a paper published in Cancers (PMID; 32326249) and a paper written by the author of this article (PMID; 31847695). This review paper itself appears to be well written. The main problem, however, is that it is not clear what is new in the review article compared to many other recently published review articles, including their own. Therefore, unfortunately, this review paper does not seem to be worthy of publication in Cancers.”
  • Response:
  • We would like to thank this reviewer for his effort in reading our manuscript and comparing it to others. The reviewer mentioned that two recent articles have been published discussing the role of lncRNAs in OC and that our article does not provide significant new data. However, our article focuses on discussing the role of lncRNAs in the development of resistance to platinum- and taxane-based chemotherapy. Furthermore, we offer significant new perspectives and insights on the interplay between lncRNAs and the molecular circuitries implicated in chemoresistance to determine their impacts on therapeutic response. We classified the lncRNAs based on the mechanistic pathway underlying chemoresistance. Thus, the major aim of the current article is based on how lncRNAs could participate in the development of acquired chemoresistance in OC.
  • On the other hand, the two articles that the reviewer#2 provided as examples differ completely from ours. For instance, the first one entitled “The Challenges and Opportunities of LncRNAs in Ovarian Cancer Research and Clinical Use”, PMID: 32326249, highlights the importance of lncRNAs in diagnosis, prognosis and treatment choice from a general perspective. Also, they review the effects of lncRNAs associated with OC in the context of cancer hallmarks. Furthermore, they discuss the molecular mechanisms by which lncRNAs become involved in cellular physiology; the onset, development and progression of OC; and lncRNAs’ regulatory mechanisms at the transcriptional, and post-translational stages of gene expression. Finally, they compile a series of online resources useful for the study of lncRNAs, particularly in the context of ovarian cancer. Thus, we think that the major aim of this article are totally different in that we provide a detailed discussion for the molecular mechanisms implicated in the development of chemoresistance to standard chemotherapy in OC, while the authors of this article provide a generalized overview for lncRNAs in OC and discuss the mechanisms of resistance in only two paragraphs.
  • Also, this reviewer mentioned a second manuscript entitled “Long non-coding RNAs in ovarian cancer: expression profile and functional spectrum” that has been published by our group during the last year. This manuscript also highlights the various roles of lncRNAs in ovarian cancer from a general perspective regarding the mechanisms implicated in the development of the disease itself. Also, as elaborated earlier, this article does not focus on the underlying mechanisms for the development of ovarian cancer. Importantly, our article not only discusses the mechanisms underlying the development of chemoresistance, but also provides a five explanatory figures and two tables that summarize the potential role of lncRNAs in chemoresistance in terms of molecular mechanisms.

Reviewer 3 Report

The authors have written a comprehensive review on the potential roles of lncRNA in promoting chemoresistance in ovarian tumors. The manuscript is generally well written and focused. The goal is to provide information of the roles of lncRNA in promoting chemoresistance against platinum and taxol agents. Appropriate references have been provided and the background information given to introduce the concepts is also appropriate and accurate. 

The only critique is that some description of the challenges in studying lncRNAs and their biological roles should have been provided. 

Authors should also describe some of the controversies in the field. For example a recent study in zebrafish where the role of lncRNA was shown to be redundant. The authors should comment on how such redundant mechanisms could be in play also in cancer chemoresistance setting. 

Author Response

  • Reviewer #3 Comment:
    The authors have written a comprehensive review on the potential roles of lncRNA in promoting chemoresistance in ovarian tumors. The manuscript is generally well written and focused. The goal is to provide information of the roles of lncRNA in promoting chemoresistance against platinum and taxol agents. Appropriate references have been provided and the background information given to introduce the concepts is also appropriate and accurate. The only critique is that some description of the challenges in studying lncRNAs and their biological roles should have been provided. Authors should also describe some of the controversies in the field. For example a recent study in zebrafish where the role of lncRNA was shown to be redundant. The authors should comment on how such redundant mechanisms could be in play also in cancer chemoresistance setting.

  • Response:

We would like to thank this reviewer for his valuable comments. By reviewing the comments, we found that it is plausible to add a section highlighting the challenges for the application of lncRNAs. Therefore, we added a section regarding the major challenges of lncRNAs. Please refer to line 683 – 767.
Also, we addressed the reviewer comment about the redundant effect of lncRNAs via citing zebrafish study and commenting on such discrepancy by raising the low conservative expression of lncRNAs across different species. Please refer to lines 725 – 745.

Reviewer 4 Report

This manuscript is an excellent overview of the current knowledge on the role of lncRNAs in ovarian cancer chemoresistance. 

Major part of the manuscript is focused on the defining various mechanisms of lncRNAs in chemoresistance. Authors have provided a detailed summary of the studies which describes various roles/mechanisms of lncRNAs in chemoresistance, primarily platinum and taxane resistance. 

The figures and tables provided in the manuscript are high quality and well summarized. Authors have further discussed the therapeutic potential of lncRNAs. The limitations discussed by the authors in the conclusions section further enhances the overall quality of this manuscript. This shows the unbiased and practical approach by the authors. 

The manuscript is well written and well structured. It doesn't require any major changes however there is one aspect that authors could discuss to further enhance the quality of the manuscript:

  • Although the technological limitations and lack of in-vivo validations is discussed by the authors however, it would be great to further elaborate on the overall challenges of gene therapy itself, if that is what is proposed by the authors as a therapeutic approach. Also, what other methods authors would propose by which a practical therapeutic strategy could be achieved? 
  • Instead therapeutic, can lncRNAs be use as diagnostic markers? This could be discussed. Perhaps as diagnostic markers lncRNAs could be useful in detection or monitoring the prognosis? is the detection limit of lncRNAs (statistically significant distinction between lncRNA from cancer cells vs normal cells) been achieved? considering the instability of RNAs in blood, what would be an ideal method of detection in patients (exosomes - RNA is well protected in exosomes, but exosome detection too have limitations)? 
  • What in the authors opinion would be next steps to achieve practical usage of lncRNAs in clinics? 

Overall, the manuscript is a good source of information, well summarized studies, and easy to read. 

Author Response

  • Reviewer#4 comment#1:

“Although the technological limitations and lack of in-vivo validations is discussed by the authors however, it would be great to further elaborate on the overall challenges of gene therapy itself, if that is what is proposed by the authors as a therapeutic approach.”

  • Response #1
  • We appreciate the comments provided by this reviewer since we think it is a great opportunity to address them in our manuscript. We added a section discussing the challenges against lncRNAs either related to intrinsic features of lncRNAs or associated with lncRNA-targeting approaches. Please refer to line 683 – 767.

  • Reviewer#4 comment#2:

“Also, what other methods authors would propose by which a practical therapeutic strategy could be achieved?”

  • Response #2

We added a paragraph highlighting the use of siRNA DOPC, a neutral nanoliposomal formulation, in clinical trials for targeting EphA2 in OC and other solid tumors. We propose that, it is plausible to use this approach for selectively delivering lncRNA-targeting siRNAs to cancer cells while minimizing systemic toxicity. Also, the use of BC-819 plasmid vector could be of valuable importance as lncRNA-based therapeutic strategy. Please refer to lines 7516 – 767 and 793 – 805.

  • Reviewer#4 comment#3:

“Instead therapeutic, can lncRNAs be use as diagnostic markers? This could be discussed. Perhaps as diagnostic markers lncRNAs could be useful in detection or monitoring the prognosis? is the detection limit of lncRNAs (statistically significant distinction between lncRNA from cancer cells vs normal cells) been achieved? considering the instability of RNAs in blood, what would be an ideal method of detection in patients (exosomes - RNA is well protected in exosomes, but exosome detection too have limitations)?”

  • Response #3

We have highlighted the potential role of lncRNAs as diagnostic or prognostic biomarkers in OC. Please refer to section 7, “lncRNAs as diagnostic and prognostic biomarkers”. Also the reviewer is asking “is the detection limit statistically significant to distinguish between cancer and normal cells? Several reports indicated that certain lncRNAs are aberrantly expressed in OC patients and such dysregulated expression correlates with clinicopathological parameters including metastasis, histological type, and tumor size. Please refer to lines 650 – 674. Furthermore, we think that for lncRNAs to be used as biomarkers, several characteristics should be identified including that lncRNAs should have a statistically significant expression in cancer vs normal patients. Please refer to lines 775 – 792.

Furthermore, the reviewer raises a major challenge for using lncRNAs as biomarkers by stating that “considering the instability of RNAs in blood, what would be an ideal method of detection in patients (exosomes - RNA is well protected in exosomes, but exosome detection too have limitations)? We addressed this comment by summarizing the challenge for lncRNAs instability in plasma and how exosomal lncRNAs could be used in this setting. Please refer to lines 702 – 724.

  • Reviewer#4 comment#4:

“What in the authors opinion would be next steps to achieve practical usage of lncRNAs in clinics? Overall, the manuscript is a good source of information, well summarized studies, and easy to read.”

  • Response #4

We addressed this point by suggesting several approaches that can be used to achieve practical use of lncRNA-based therapeutics. Please refer to lines 805 – 813. Also, several characteristics are required for lncRNAs to be used as successful diagnostic and/or prognostic biomarkers. Please refer to line 775 – 805 and line 815 – 823.

Round 2

Reviewer 2 Report

I understand the argument of the authors. There is value in this review paper.